# Structural characterisation of the *Chaetomium thermophilum* Chl1 helicase

Zuzana Hodáková[1,2¤]*, Andrea Nans[3], Simone Kunzelmann[3], Shahid Mehmood[4], Ian Taylor[5], Frank Uhlmann[6], Peter Cherepanov[2], Martin R. Singleton[1]*

**1** Structural Biology of Chromosome Segregation Laboratory, The Francis Crick Institute, London, United Kingdom, **2** Chromatin Structure and Mobile DNA Laboratory, The Francis Crick Institute, London, United Kingdom, **3** Structural Biology Science Technology Platform, The Francis Crick Institute, London, United Kingdom, **4** Proteomics Science Technology Platform, The Francis Crick Institute, London, United Kingdom, **5** Macromolecular Structure Laboratory, The Francis Crick Institute, London, United Kingdom, **6** Chromosome Segregation Laboratory, The Francis Crick Institute, London, United Kingdom

¤ Current address: Research Institute of Molecular Pathology, Vienna BioCenter, Vienna, Austria
* Martin.Singleton@crick.ac.uk (MRS); Zuzana.Hodakova@crick.ac.uk (ZH)

## Abstract

Chl1 is a member of the XPD family of 5'-3' DNA helicases, which perform a variety of roles in genome maintenance and transmission. They possess a variety of unique structural features, including the presence of a highly variable, partially-ordered insertion in the helicase domain 1. Chl1 has been shown to be required for chromosome segregation in yeast due to its role in the formation of persistent chromosome cohesion during S-phase. Here we present structural and biochemical data to show that Chl1 has the same overall domain organisation as other members of the XPD family, but with some conformational alterations. We also present data suggesting the insert domain in Chl1 regulates its DNA binding.

## Introduction

Sister chromatid cohesion (SCC) is a key process to ensure equal distribution of genetic material to daughter cells during cell division. Central to this process is the cohesin complex; a ring-shaped macromolecular assembly which topologically embraces DNA [1]. Cohesin is deposited on chromatin by the cohesin loader in G1 and, immediately from their replication in S-phase, holds the sister chromatids together until mitosis [2, 3]. The persistent establishment of cohesion occurring during replication requires acetylation of the Smc3 cohesin subunit by Eco1 acetyltransferase [4, 5]. Additional cohesion establishment factors include the fork protection complex (FPC); a three-protein complex comprised of Tof1, Mrc1 and Csm3 [6]; the PCNA clamp loader RFC[Ctf18] [7]; the Ctf4 trimer [8]; and the Chl1 helicase [9]. Genetic alterations in individual proteins are not lethal in yeast, but combinations of their deletions result in a loss of viability associated with severe cohesion defects [7].

The Chl1 helicase was the first chromosome loss mutant identified in budding yeast [10]. Chl1 and Ctf4 were found to interact during DNA replication [11]. Their deletion in Eco1-lacking cells is synthetic lethal in yeast, suggesting that these two proteins act in parallel to the Eco1 cohesin acetylation pathway [12]. Ctf4, a stable component of the replisome, recruits

**Data Availability Statement:** All EM volume files are available from the EMDB database, accession number EMD-12817.

**Funding:** MRS was supported by the Francis Crick Institute, which receives its core funding from Cancer Research UK (FC001155), the Medical Research Council (FC001155) and the Wellcome Trust (FC001155). cancerresearchuk.org wellcome.org ukri.org The funders had no role in study design, data collection and analysis, decision to publish, or preparation of the manuscript.

**Competing interests:** The authors have declared that no competing interests exist.

Chl1 via a conserved Ctf4-interacting peptide box with the sequence "DDIL" shared with other proteins including Sld5 of GINS, a component of the main replicative helicase CMG, and Pol1 of the Pol α-primase [11, 13]. Their interaction is independent of Chl1 helicase activity, as no disruption in binding and only minor cohesion defects were observable in the K48R ATPase-dead Chl1 mutant [11]. The mammalian Chl1 ortholog, DDX11, utilises its helicase activity for processing guanine quadruplex impediments ahead of the replication fork to prevent replisome stalling. This function is dependent on the interaction with Timeless, the human homolog of yeast Tof1 [14]. Cryo-electron microscopy (cryo-EM) structure of the yeast CMG-FPC (Cdc45-MCM-GINS-FPC) have revealed that Tof1 is positioned ahead of the replisome in direct contact with DNA to provide a stronger grip for subsequent CMG helicase processing [15]. In higher eukaryotes, DDX11 and Timeless together resolve impediments ahead of the replisome, as well as mediate SCC by a yet not fully understood mechanism [16, 17]. The helicase activity of DDX11 is essential for both functions, and Timeless apparently enhances ATPase and DNA-binding activities of DDX11 [18]. This is consistent with observations from studying the cellular phenotypes of patients with the Warsaw breakage syndrome, a hereditary disease caused by mutations in DDX11, where patients display premature sister chromatid separation and a higher sensitivity to crosslinking reagents and ultraviolet light resulting in DNA damage [19, 20]. Whereas the role of Chl1/DDX11 in SCC has been described in both yeast and mammalian cells, information about its role in replication stress response has not been elucidated in yeast. The primary interaction in yeast seems to occur between Chl1 and Ctf4, although Ctf4 does not modulate Chl1's activity [11].

Chl1 belongs to the XPD subfamily of helicases, together with Rtel1, FancJ and XPD, all of which are implicated in genome maintenance. They are predicted to share an overall architecture featuring two helicase domains separated by the so-called Arch domain [21, 22]. These 5'-3' processing helicases are further characterised by a conserved 4Fe-4S iron sulphur (Fe-S) cluster, ligated by three conserved and one variable cysteine residues. The cluster is involved in DNA unwinding in concert with the Arch domain [23]. The crystal structure of ssDNA-bound DinG, the bacterial XPD homolog, revealed that DNA sits in a positively charged tunnel formed by the helicase domains and the Arch domain, where the bases are stacked onto one another until reaching the P-motif; a canonical motif which causes the bases to flip out of the stacked conformation. ATP binding and hydrolysis-driven conformational changes in the helicase domains lead to strand translocation [24]. The conservation of both the canonical motifs between the XPD helicases and the predicted similarities in architecture suggest a shared mechanism of action.

Here we describe a 7.7Å cryo-EM structure of the *Chaetomium thermophilum* Chl1 protein confirming that XPD helicases share a similarity in their domain organisation. Compared to XPD, Chl1 proteins contain an additional 20-kDa domain inserted in its helicase domain 1 (HD1), which can potentially mediate availability of the DNA-binding tunnel and thus the helicase activity of the protein.

## Results

### Chl1 is a monomeric protein

Chl1 belongs to the Fe-S containing XPD subfamily of helicases, 5'-3'-unwinding proteins which are predicted to share a similar domain architecture (Fig 1A). We chose to study Chl1 from the thermophilic fungus *Chaetomium thermophilum*. Chl1 shares an approximate 37% sequence identity with budding yeast (*Sc*Chl1*)* and human (*Hs*Chl1) proteins, both of which were tested for expression (not shown) but resulted in very low expression levels. Conversely, the *Chaetomium* Chl1 showed a high level of expression and was thus chosen for these studies

**A**

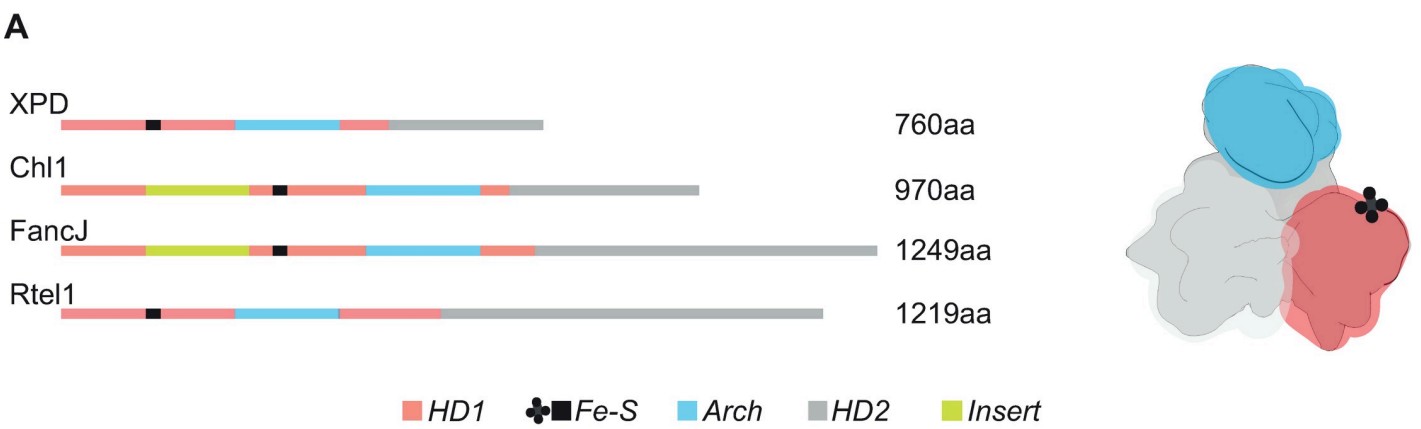

**B**

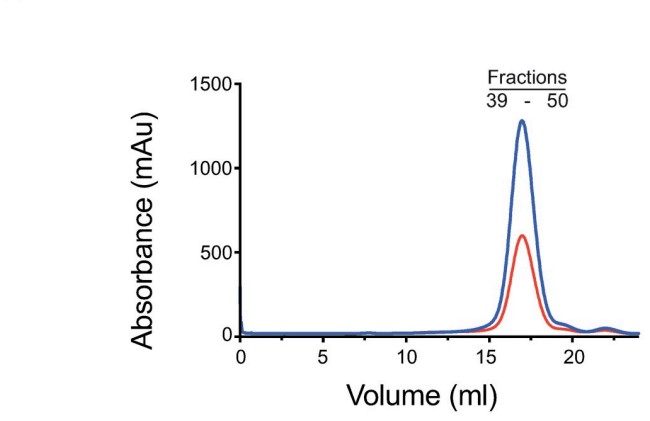

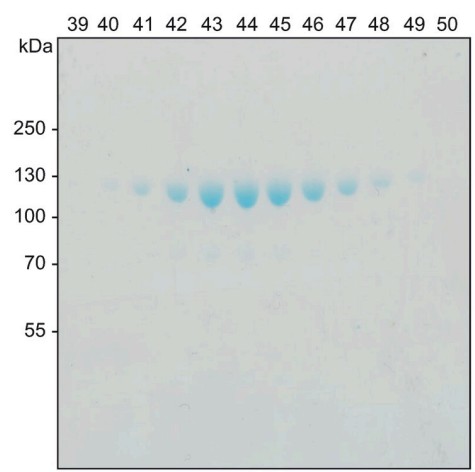

**C**

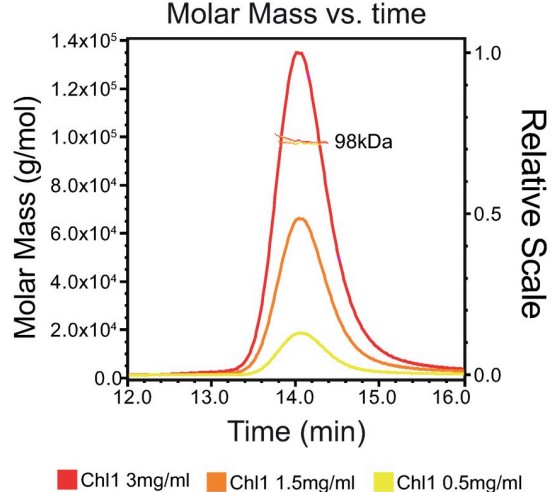

**D**

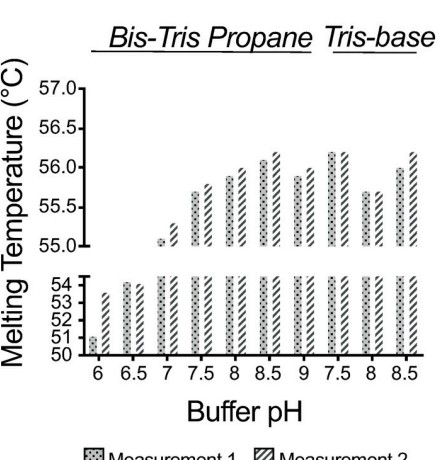

**Fig 1. Characterisation of Chl1. (A)** Schematic representation of the sequences and generalised structure of the human XPD subfamily helicases. Locations of functional domains are identified by colour. **(B)** Size exclusion chromatography elution profile and SDS-PAGE analysis. **(C)** SEC-MALS at three working concentrations. **(D)** Thermal stability assays.

for both the high obtainable protein yields and favourable thermostable properties. The similarity of function to the yeast protein is inferred from homology, although this has yet to be experimentally tested. The codon-optimised gene for Chl1 was cloned into pFastBac vector suitable for insect cell expression system and purified using a two-step purification including affinity and size exclusion chromatography (Fig 1B). The purified protein had a yellow colour as a result of the presence of the Fe-S cluster, observed previously with purification of XPD proteins [23]. Analysis of the oligomeric state of Chl1 using size exclusion chromatography with multiangle light scattering (SEC-MALS) revealed that the protein is in a monomeric state across three concentrations used in this study (Fig 1C).

## Chl1 structurally resembles XPD

We next set out to structurally characterise the full-length Chl1 protein. We first determined that the highest stability is achieved in buffers ranging from pH 7.5–9. pH 8.5 was selected as the protein showed the highest melting temperature in thermal stability assays in a buffer of this pH (Fig 1D). Despite extensive efforts, we were unable to obtain diffracting crystals of the full-length protein, or a range of truncated constructs. Therefore, we carried out structural analysis by single-particle electron microscopy. Initial negative staining analysis revealed a well behaved and dispersed sample (see below). Given the limited achievable resolution with negative staining, Chl1 was vitrified for analysis by cryo-EM. The protein localised in the open holes of the grids but formed small aggregates which hindered particle picking. We overcame this issue by introducing a low concentration of the lauryl maltose neopentyl glycol detergent which resulted in monodispersed particles in vitrified ice (Fig 2A). A dataset yielding nearly 1.9 million particles was collected on a Titan Krios microscope operating at 300 kV (Table 1). Reference-free 2D classification was performed on particles extracted from high-pass filtered micrographs (Fig 2B, S1 Fig in S1 File). Subsequent 3D refinements converged at 7.7 Å resolution (S2 Fig in S1 File). We suspect that this resolution is an overestimate with the true resolution being slightly lower, as the features which should be starting to become visible as this resolution, such as α-helices, are only poorly visible. Furthermore, the phase-randomised curve from the FSC plot of the final reconstruction suggests some degree of over-fitting (S2C Fig in S1 File), despite our best efforts to overcome this by minimising preferred orientations and using the SideSplitter reconstruction algorithm [25]. We have therefore been cautious in our structural analysis and only consider details which are likely to be real given the limited resolution of the final reconstruction or may be seen in the 2D class averages.

The final model shows a domain composition similar to the XPD protein, confirming structure predictions and suggesting a conserved architecture of the XPD subfamily proteins [26]. Chl1 can be separated into three domains: the two helicase domains HD1 and HD2 and the Arch domain (Fig 2C). The Arch domain is clearly visible as it extends above the two helicase domains. No nucleotide or DNA was added to the grid preparation, and the $A_{280}/A_{260}$ UV absorbance ratios suggested no endogenous nucleotides had co-purified with the protein. We therefore assume that this structure represents the nucleotide-free conformation. The Chl1 conformation differs to the nucleotide-free structures of XPD, which all show an Arch domain in a so-called "closed" conformation (Fig 2D and 2E) [22, 27]. The only structures with an open conformation are those with a disrupted Fe-S cluster [21, 22]. The Fe-S cluster in Chl1 is

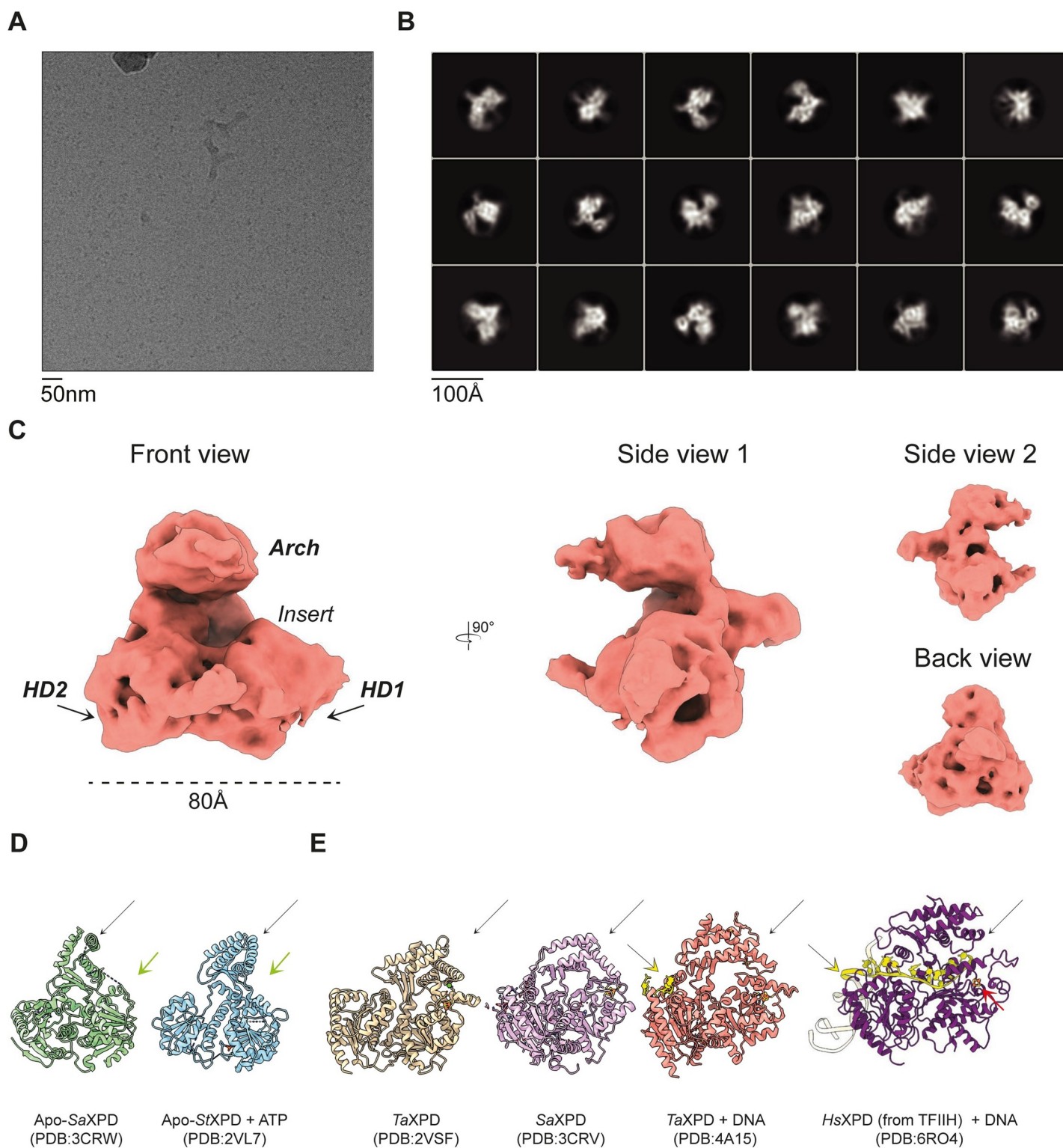

**Fig 2. The 7.7Å reconstruction of Chl1. (A)** A representative micrograph from the data collection. (**B**) Final 2D classification of 157 000 particles. (**C**) The final 3D reconstruction obtained in Relion-3 using Sidesplitter. (**D**) Open conformations of apo-XPD structures with a missing Fe-S cluster. The partially disordered Fe-S domain is annotated by green arrows. (**E**) The "closed" conformation of XPD with a folded Arch domain. The Fe-S cluster is annotated with a red arrow. DNA binding to the helicase, shown by a yellow arrow, occurs with the Arch domain in its folded state. The Arch domain in (D) and (E) are annotated by a black arrow.

**Table 1. Data collection and processing.**

| | |
|---|---|
| Magnification | 165,000 x |
| Voltage (kV) | 300 |
| Defocus range (μm) | -1.5 to -3.6 |
| Pixel size (Å) | 0.839 |
| Symmetry imposed | C1 |
| Movies | 4843 |
| Frames per movie | 10 |
| Initial particle images (no.) | 1,887,048 |
| Final particle images (no.) | 88,452 |
| Dose rate (e⁻/Å²/s) | 7.4 |
| Total exposure time (sec) | 10 |
| FSC threshold | 0.143 |
| Final map resolution (Å) | 7.7 |

Dose rate ($e^-/Å^2/s$)

likely intact as the protein preparations are yellow, suggestive of the presence of the cluster bound to the protein (Fig 3A).

## The Fe-S cluster is intact

To confirm that both the open Arch domain and the lack of strong secondary structure of the insert is not a result of the missing cluster, we performed hydrogen-deuterium exchange-mass spectrometry (HDX-MS) experiments and analysed the data for the regions containing the cysteine residues forming sulfide bonds with the Fe-S cluster (Fig 3B). Peptide coverage was almost 80% for this region which spans approximately 90 amino acids. Peptides spanning two of the three conserved cysteines were observed in the experiment, and displayed slow hydrogen-deuterium exchange rates. The remaining two cysteines' proximal regions were recovered by the digestion and also show low exchange rates. These observations suggest that the Fe-S-

**A**

**B**

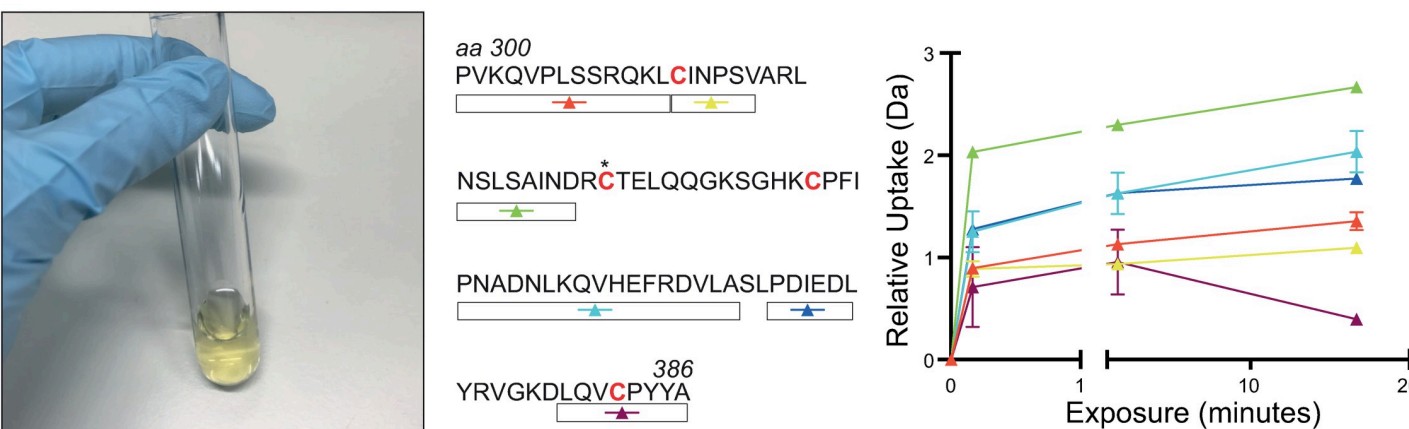

**Fig 3. Characterisation of the Fe-S cluster-binding cysteines by HDX-MS. (A)** The yellow colour of the sample from SEC elution. (**B**) HDX-MS mapping of the Fe-S Cluster. On the left, peptide coverage for the 90 amino acid region covering the four cysteines binding to the cluster is shown. The asterisk denotes the variable cysteine. H-D exchange rates are shown in the graph on the right. The relatively low rates (compare Fig 4B) are indicative of a folded structure.

binding domain is structured. Taken together, the colour of the protein and the exchange rates from HDX-MS are suggestive of an intact Fe-S cluster.

## The Chl1 insert is predominantly disordered

The Chl1 insert is predicted to lie between the Walker A motif and the Fe-S domain. Sequence analysis showed no homology to any PDB entry, and alignments suggest its presence in only one other member of the XPD subfamily, FancJ (Fig 1A). The human Chl1 ortholog, DDX11, contains an EYE motif in its insert which is important for interactions with Timeless [16]. This motif, together with the N-terminally flanking residues, are conserved amongst eukaryotes, and are predicted to lie in the insert of the Chl1 proteins (Fig 4A).

To elucidate the structure of this insert, we analysed the HDX-MS data (Fig 4B). Analysis of the hydrogen-deuterium exchange curves for the insert-derived peptides, which span between amino acids 66 and 213 of the protein, revealed that the C-terminal segment of the insert is largely disordered. Conversely, the region downstream of the Walker A motif is folded. PsiPred structure predictions (S3 Fig in S1 File) suggest the presence of a large helix as an extension from the Walker A motif, which agree with our the obtained HDX-MS data. Taken together, the data suggest that the Timeless / Tof1 recognise a feature in the unstructured part of the Chl1 insert.

Simultaneously, we expressed the insert sequence as an isolated domain (S1 Table in S1 File). Expression of inserts from five different species yielded soluble proteins (Fig 4C), suggesting that the insert can exist as a separate domain. Two inserts, from the *C. thermophilum* Chl1 and human DDX11 proteins, were successfully purified as soluble species by gel filtration, where the human insert potentially formed a dimer as well as a monomer (Fig 4D). Attempts to manipulate the boundaries of the inserts by shortening the N terminus resulted in the protein becoming insoluble (not shown). To further characterise the insert, we expressed a Chl1 construct with an internal deletion of the disordered section of the insert domain, (referred to as MiniChl1$^{V1}$), based on previously reported boundaries of the insert and the linker used [11]. Comparing this construct to the full-length Chl1 by negative staining revealed a potentially destabilised protein (Fig 5A). This protein construct did not yield clear 2D classes in either negative stain EM or Cryo-EM (not shown). The introduced heterogeneity to the protein together with the data obtained from HDX-MS led us to revisit the construct design, as the low sequence conservation in the inserts could have resulted in incorrect boundaries been used for the *Chaetomium* construct. We therefore produced a second construct with a smaller internal deletion, termed MiniChl1$^{V2}$, where only the C-terminal disordered region of the insert was removed. Negative staining analysis revealed a stable protein reminiscent of the full-length Chl1 (Fig 5A). The structural stability of the new construct and, thus, the correct boundaries of the disordered segment of the insert, are further suggested by the overlay of SEC elution profiles (Fig 5B), which showed that all three proteins elute within the same elution volumes despite MiniChl1$^{V1}$ being by 20kDa smaller. The elution profiles of the full-length protein and the Mini Chl1$^{V2}$ are suggestive of similar hydrodynamic properties despite the absence of the unstructured segment of the insert.

## Conformational rearrangements of Chl1 and DNA binding activity

In order to investigate possible structural rearrangements upon nucleotide binding, we vitrified Chl1 in the presence of nucleotide and a 22-mer ssDNA oligonucleotide. 2D classification of the particles resulted in class averages highly similar to those obtained with DNA- and nucleotide-free samples (Fig 6A and 6B). Although at this resolution, it was not possible to observe nucleotide-induced motion of the helicase domains, the Arch domain is clearly visible

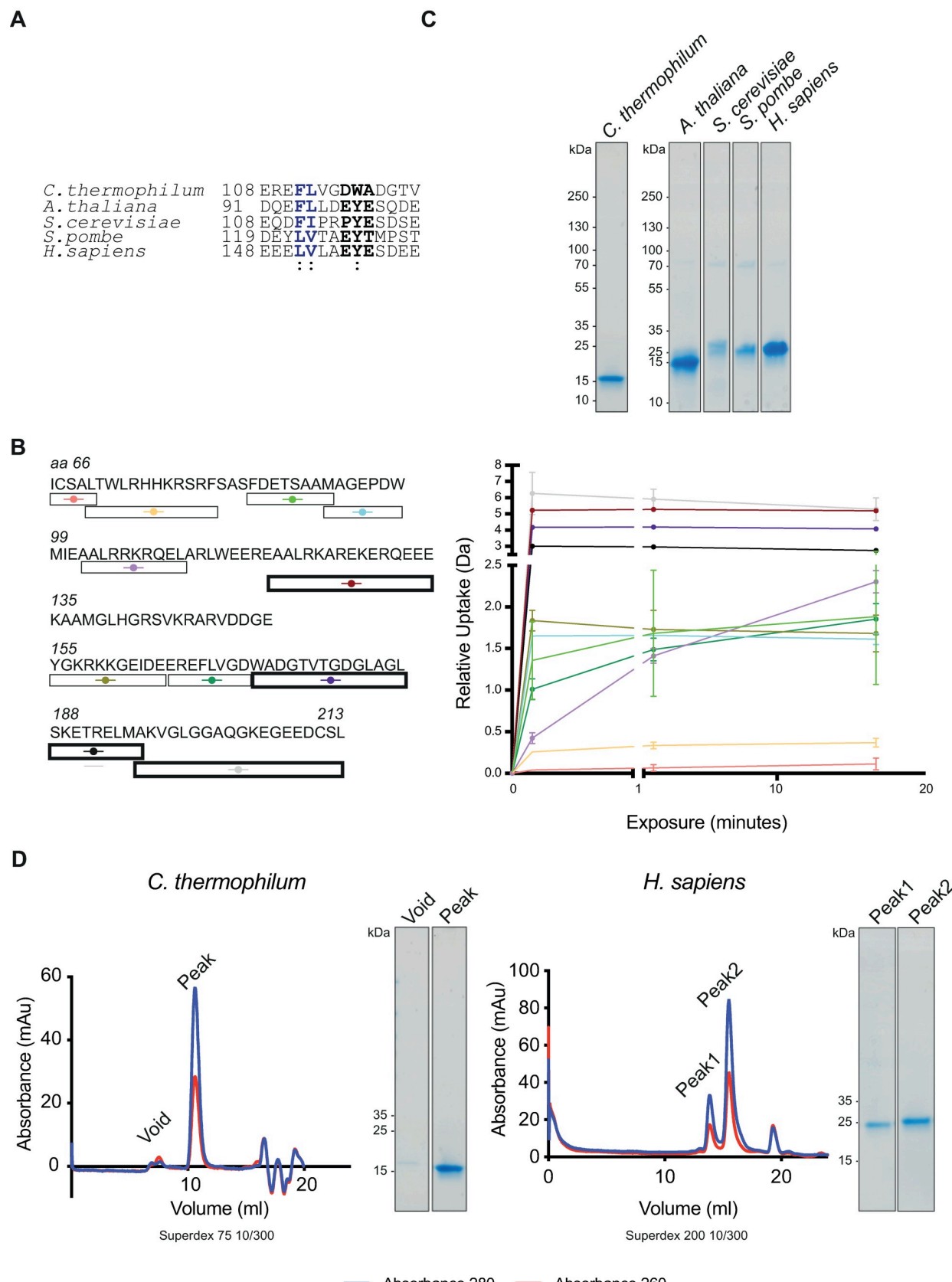

**Fig 4. The Chl1 insert.** (**A**) Conservation of the Tof1-binding motif (black) and the flanking hydrophobic residues (blue) in the species expressed in (C). (**B**) Peptide coverage for the insert amino acid region (residues 66–213). H-D exchange rates for the open boxes are shown in the graph on the right. Highlighted boxes represent sequences with the highest exchange rates. (**C**) Expression of the insert from five indicated species. The borders of these inserts were selected primarily based on secondary structure predictions. (**D**) SEC elution profiles of two inserts.

above the helicase domains in an "open" conformation. DNA-bound structures of XPD showed a closed conformation of this domain (Fig 2E) [24, 28, 29]. Since the presence of an intact Fe-S cluster is suggested by our HDX-MS data, the open arch domain conformation is thus potentially biologically relevant, and is not an artifact caused by the Fe-S cluster disruption. We further attempted to characterise the DNA-binding activity of Chl1 by fluorescence anisotropy spectroscopy. A 22-mer ssDNA oligonucleotide with a 5'-fluorescein tag was

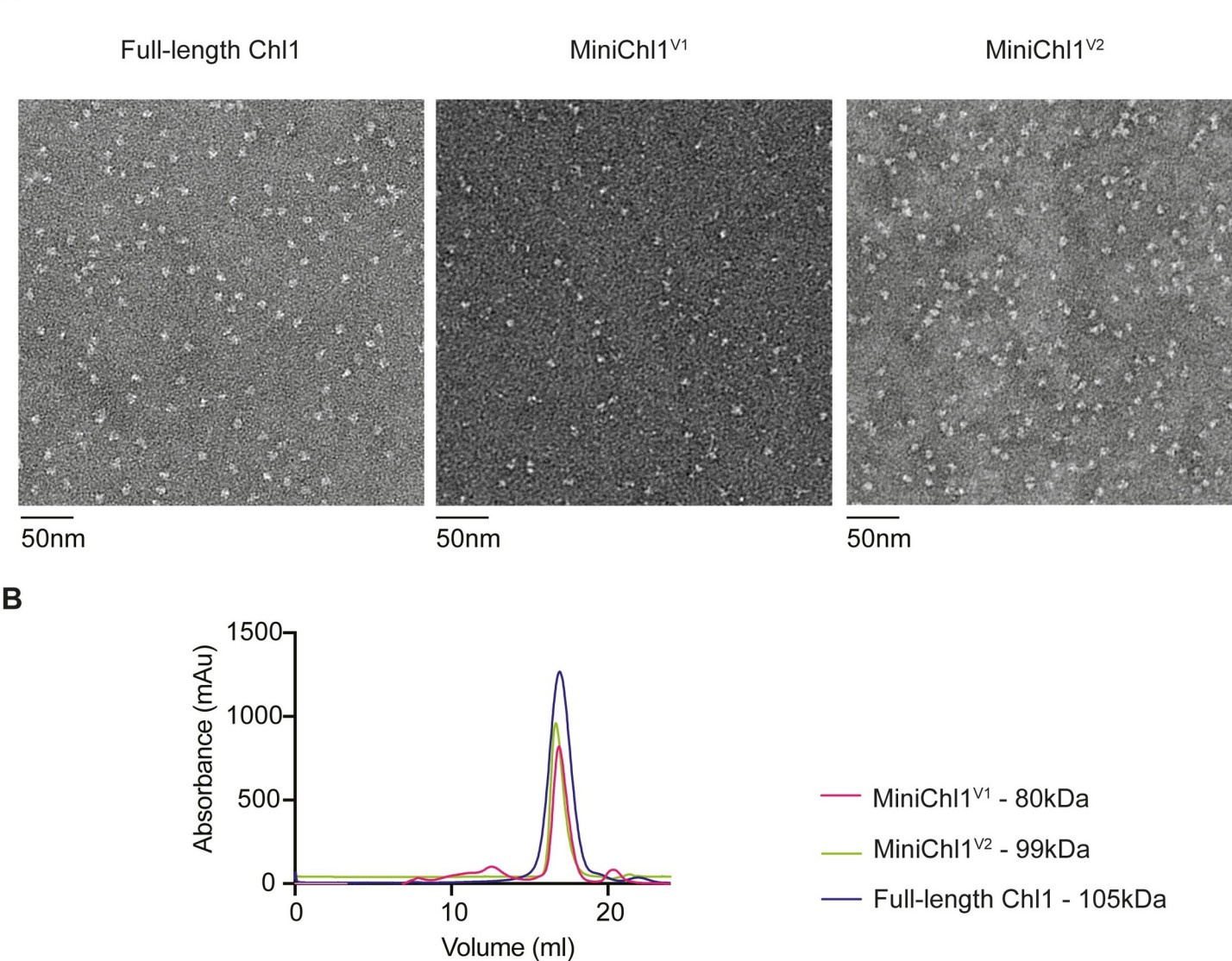

**Fig 5. Deletion of the insert of Chl1.** (**A**) A comparison of the full-length Chl1 protein and two internal deletions of the insert to create MiniChl1[V1] and MiniChl1[V2], respectively, using negative staining. Note poor particle quality in the MiniChl1[V1] construct. (**B**) An overlay of the elution profiles of all three proteins, and their molecular weights.

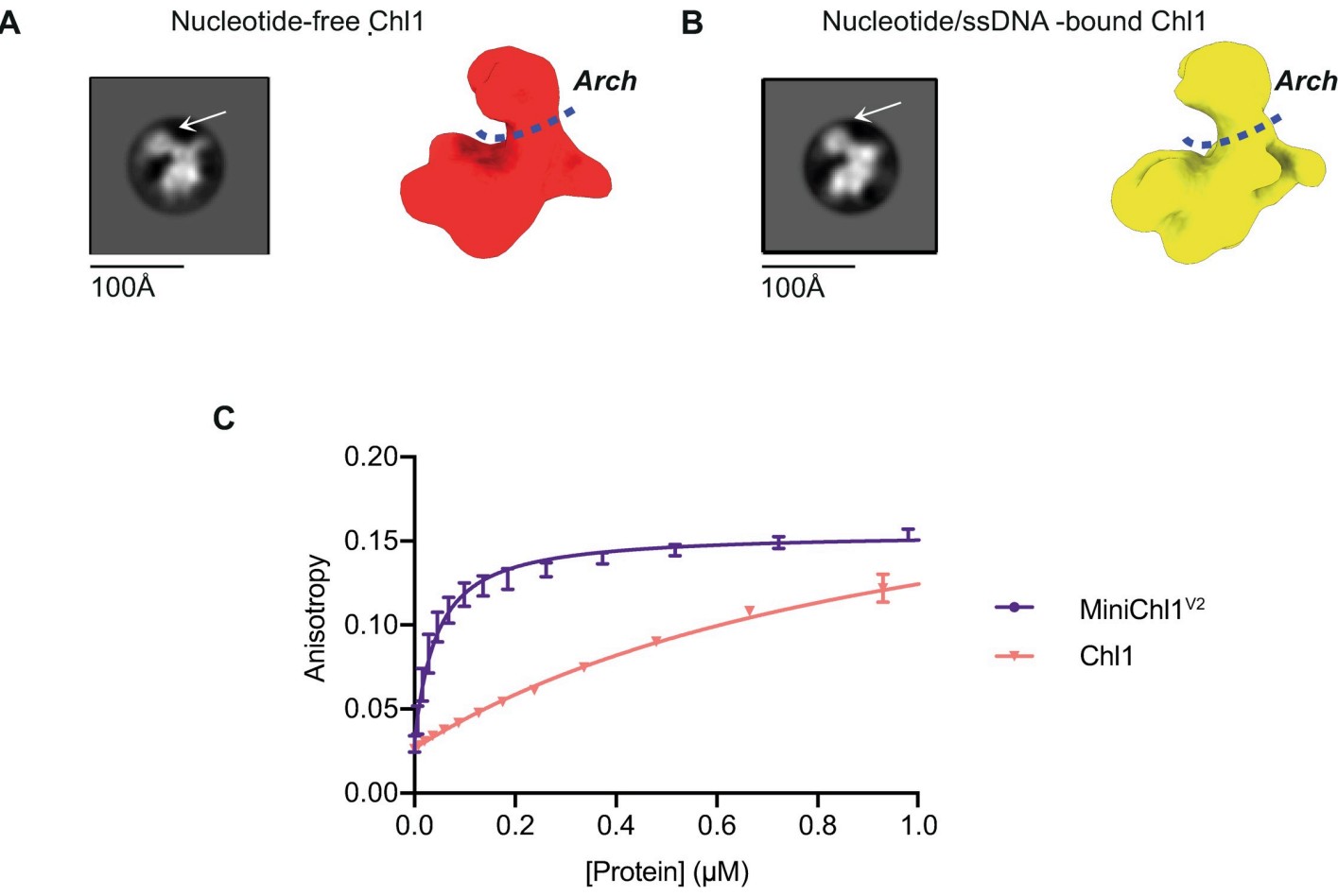

**Fig 6. Changes in protein conformation which lead to DNA binding.** Highest populated 2D class and a final 3D volume of (**A**) the nucleotide-free Chl1 and (**B**) the nucleotide and ssDNA-supplemented Chl1. For both datasets the same collection parameters were used. (**C**) Binding curves of ssDNA to full-length and truncated Chl1 samples.

incubated with full-length Chl1 but no plateau was reached in the binding curve, suggesting low ($> 1\mu M$) affinity (Fig 6C). However, removal of the insert domain and repeating the experiment with MiniChl1$^{V2}$ allowed us to measure the DNA-binding affinity with an estimated $K_d$ of 38 nM.

## Discussion

Here we present, to our knowledge, the first cryo-EM structure of Chl1. As the protein is refractory to crystallisation, and relatively small by EM standards, we were only able to obtain a reconstruction of limited resolution. Nevertheless, the structural data and biochemical analyses allow us to draw several conclusions. The structure of Chl1 shows a high resemblance to its homolog, XPD as expected from sequence conservation. Interestingly, unlike XPD, the Arch domain in the Chl1 structure adopts an open conformation not folded towards the helicase domains. An open conformation of the Arch domain has previously only been observed for XPD structures which are in their Apo state; with a disrupted Fe-S cluster [21, 22]. The combination of biochemical and mass spectrometry analysis shows that our Chl1 sample contains an intact cluster as the regions surrounding the cluster adopt a secondary structure, which would be unlikely with a disrupted cluster. Chl1 contains four cysteines which hold the cluster, with

three of these being absolutely essential for protein function [23]. Despite the lack of coverage for two cysteines, the peptide coverage does span to surrounding regions. Soaking saXPD crystals in ferricyanide-containing cryosolution for cluster removal resulted in a complete disorder of these regions [22]. The open conformation of the Chl1 Arch domain is therefore not a result of the Fe-S cluster disruption.

The Arch domain is important in DNA translocation, working together with the Fe-S cluster to unwind DNA in a 5'-3' direction. Its movement has been determined by crystallising the DinG bacterial homolog of XPD in multiple catalytic states [24]. Even after ATP hydrolysis, the Arch domain remains in a closed state. We did not observe a change in the Arch domain position between nucleotide-free or nucleotide-supplemented samples. We do not expect this observation to be caused by the domain's flexibility based on the local resolution estimations as it appears to be the best-resolved part of the protein. The Arch domain is therefore in a stable conformation, further supported by being a predominant feature visible in 2D class averages.

Within the core helicase domains, HD1 displays lower local resolution than HD2. HD1 contains the partially disordered insert and the predicted region where this insert lies shows one of the lowest local resolutions throughout the whole reconstruction. The core of HD1, responsible for performing ATP binding and hydrolysis, reaches a higher resolution than the insert. The helicase domain is therefore most likely in a relatively static conformation, and the lower resolution of the insert is not due to a movement of this helicase domain.

Our data show that full-length Chl1 has very low affinity for DNA. This is surprising, as the primary requirement for helicase activity is DNA engagement. Strikingly, removal of the Chl1 insert results in a significant increase in DNA binding, resulting in an affinity very similar to that of XPD (46 nM) [22]. The differing position of the Arch domain relative to the XPD structures raises the question of whether this domain performs a similar function in Chl1 than it does in XPD and potentially in the remaining helicases in this family, which remain structurally uncharacterised. In eukaryotic organisms, XPD is a part of the TFIIH where it performs roles in transcription and nucleotide excision repair, only the latter being helicase activity-dependent [30]. XPD must first be relieved from its auto-inhibition by the release of the MAT1 inhibitory subunit of CAK and activation by XPA [31]. The Arch domain is responsible for this auto-inhibition; the so-called "plug" segment of the Arch domain inserts itself into the DNA-binding cleft [32, 33]. Only upon structural rearrangements caused by XPA does the plug move away from the DNA-binding site and DNA can engage [34]. Sequence alignments of human XPD subfamily proteins shows that this negatively-charged plug segment is present in all proteins but Chl1. Instead, Chl1 contains a different negatively charged segment which could block DNA binding: the insert, having a net negative charge. Interestingly, FancJ contains both the plug domain and the insert, where the insert mediates interactions with the MLH-1 protein of the MutLα complex involved in helicase function-dependent interstrand crosslink resolution [35]. The plug domain of FancJ is, like in XPD, negatively charged but conversely to Chl1, FancJ's insert is positively charged. The charge conservation is most likely an important feature for inhibition, as the residues contacting the plug and DNA are the same. Therefore, a positively charged insert of FancJ most likely does not mediate DNA binding inhibition. Chl1 contains no such plug segment in its Arch domain but contains the negatively-charged insert. We therefore speculate that this insert is inhibiting DNA binding to Chl1 (Fig 7), corroborated by our findings that deleting these residues substantially increases the affinity of DNA binding to Chl1. It remains to be determined what structural rearrangements the Chl1 insert might undergo upon binding to interacting partners at the replication fork, such as Tof1 whose binding motif appears to be present in the Chl1 insert. Structurally characterising this interaction could provide new insights into the dual role of the Chl1 helicase in SCC and replication stress response.

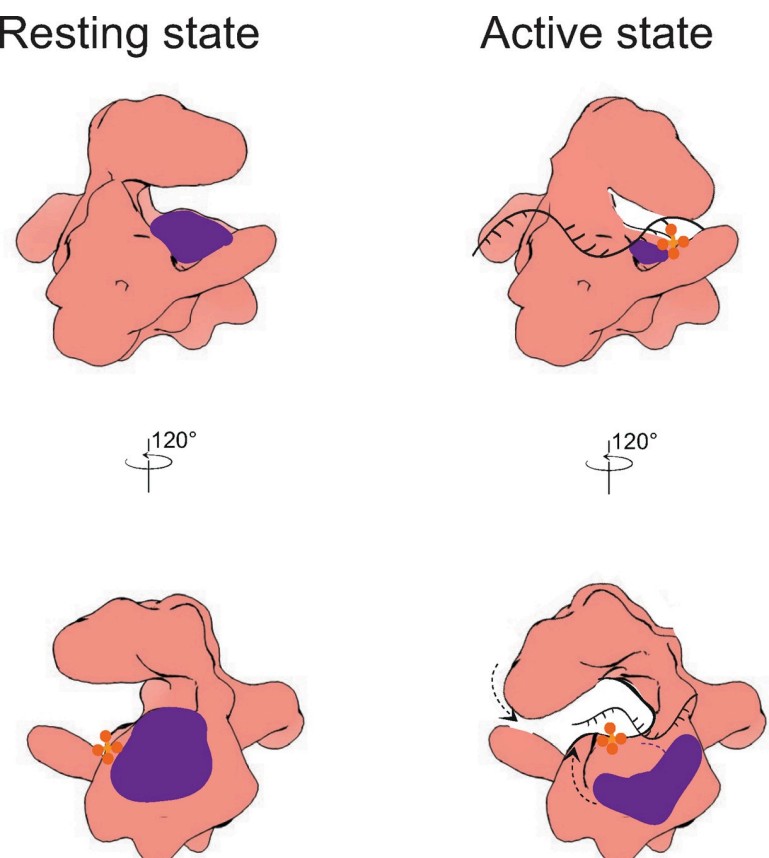

**Fig 7. A speculative mechanism of Chl1 auto-inhibition.** In its inhibited state, access to the DNA-binding site is restricted by the insert (purple) sitting in HD1. By binding of an interacting partner, structural rearrangements within the protein lead to movement of the insert away from the DNA binding cleft. As a consequence, DNA can be engaged and the Arch domain can adopt a more closed conformation.

## Methods

### Recombinant plasmids and protein expression

Synthetic genes codon-optimised for expression in *Spodoptera frugiperda* (*Sf*) insect cell line or *Escherichia coli* (*E. coli*) were purchased from GeneArt and cloned into the pFastBac vector containing an N-terminal double strep tag for insect cell expression or a modified pET28a with an N-terminal 6xHistidine tag. All genes used were *Ct* genes unless otherwise stated. Chl1 and MiniChl1 constructs were expressed in insect cells, Chl1 insert domain constructs were expressed in *E. coli*. For insect cell expression, pFastBac vectors were recombined into DH10BacY cells for bacmid production. P1 baculovirus was produced by transfecting the bacmid into *Sf*9 cells, followed by baculovirus amplification. P3 virus was used for protein expression in *Sf*9 cells, carried out at 27˚C for 3 days shaking at 130 rpm. For bacterial expression, DNA constructs were introduced in *Escherichia coli* BL21(DE3) strain. Transformed bacteria were grown in LB medium at 37˚C, and protein expression was induced by addition of 0.2 mM IPTG when $OD_{600}$ of the culture cells reached 0.6.

### Purification

Chl1 was purified from insect cells harvested in lysis buffer (300 mM NaCl, 50 mM Tris-HCl pH 8.5, 0.5 mM tris(2-carboxyethyl)phosphine (TCEP) complemented with Basemuncher

benzonase (Expedeon) and cOmplete protease inhibitor cocktail (Roche). Resuspended cells were disrupted by sonication and centrifuged at 30,000 x *g* for 45 min at 4˚C. The supernatant was loaded on a StrepTactin column (Qiagen) pre-equilibrated with lysis buffer (300 mM NaCl, 50 mM Tris-HCl pH 8.5, 0.5 mM TCEP), extensively washed with lysis buffer, and the bound protein eluted with lysis buffer supplemented with 2.5 mM desthiobiotin. The protein was further purified by size exclusion chromatography through a Superose 6 10/100 column (GE Healthcare) equilibrated in SEC buffer (150 mM NaCl, 50 mM Tris-HCl pH 8.5, 0.5 mM TCEP). For MiniChl1 constructs, an additional ion exchange chromatography step was introduced prior to size exclusion chromatography. The protein was injected into a HiTrap S column (GE Healthcare) pre-equilibrated with buffer A (100 mM NaCl, 50 mM BisTris-HCl pH 6, 0.5 mM TCEP) and eluted with a shallow 0.1–1 M NaCl gradient in buffer A.

Insert constructs were purified from *E. coli* cells harvested in lysis buffer supplemented with DNase I, protease inhibitors and 20 mM imidazole. The supernatant was loaded on a HisTrap FF column (GE Healthcare) and eluted with a shallow gradient of lysis buffer and lysis buffer supplemented with 500 mM imidazole.

## Negative staining

For negative staining, all proteins were taken from the elution peak of the size exclusion chromatography column and diluted to 0.1 μM. Copper 400-mesh carbon-coated grids (EM resolutions) were glow-discharged carbon side facing up for 30 sec at 45 mA. 4 μl protein was applied to the glow-discharged carbon for 60 sec. Most of protein solution was then blotted away and grids were stained by immersion into 4 drops of 2% uranyl acetate in a sequential manner. Excess negative stain was blotted away, and the grids were air-dried.

## Cryo-EM grid preparation

Chl1 sample was taken from the elution peak after SEC and protein concentration was adjusted to 2 μM. The sample was supplemented with 0.0003% lauryl maltose neopentyl glycol prior to vitrification. Non-glow discharged fresh C-flats 1.2/1.3 Au (EM Resolutions) were used for vitrification with Vitrobot mark IV (Thermo Fisher Scientific) at room temperature and 95% humidity. Sample was applied for 60 sec and blotted with blot force -1 for 2.5 sec prior to vitrification by plunging into liquid ethane.

## Data collection and image processing

Initial grid screening and data collection was performed on 200 kV Talos Arctica microscope equipped with a Falcon III detector operating in linear mode. Images were collected at a pixel size of 1.26 with a cumulative dose of 85 e⁻/Å$^2$ in a total of 10 frames. Subsequent data collections were performed on a 300 kV Titan Krios equipped with a post-column energy filter and a K2 Summit direct electron detector (Gatan), operating in counting mode. A defocus range of -1.5 to -3.6 μM was used during collection. Images were collected at a pixel size of 0.839 Å with a cumulative dose of 74 e⁻/Å$^2$ in a total of 40 frames. For all cryo-EM datasets, the frames were aligned and summed with dose-weighting applied using MotionCor2 [36], and subsequent CTF estimation performed using GCTF [37]. Particles were semi-automatically picked using EMAN2 [38] followed by initial 2D classification. Resulting class averages, low-pass filtered to 20 Å, were used for template-based picking with Gautomatch (http://www.mrc-lmb.cam.ac.uk/kzhang/). Following automated particle picking, a high-pass filter of 140 Å was applied and particles were subjected to reference-free 2D classification in CryoSparc-2 [39]. Initial model was generated using CryoSparc-2, followed by 3D classification and refinement using Relion-

3.1 [40, 41]. The final refinement and model was generated by Relion AutoRefine procedure implementing Sidesplitter [25]. 3DFSC was calculated in CryoSparc-2 [42].

## Hydrogen deuterium exchange

HDX-MS was performed using a Waters HDX manager. Samples were prepared by 10-fold dilutions from 5 μM Chl1 protein in deuterated or nondeuterated buffer containing 150 mM NaCl, 50 mM HEPES pH 7.5, 0.5 mM TCEP. The pH of the sample was reduced to 2.3 and an in-line pepsin-immobilized column at 20˚C was used for protein digestion. For labelling experiments, apo protein was incubated for 10 s, 100 s, and 1000 s at room temperature. All HDX-MS experiments were performed in triplicate. Sequence coverage and deuterium uptake were analysed by using ProteinLynx Global Server (Waters) and DynamX (Waters) programs, respectively. For mass correction, Leucine enkephalin at a continuous flow rate of 5 μl min$^{-1}$ was sprayed. All sample preparation and sample loading performed manually.

## SEC-MALS

SEC-MALS was performed using a Superdex 200 10/300 column (GE Healthcare) pre-equilibrated with SEC buffer supplemented with 0.05% sodium azide, followed by sample injection into the Dawn8+ MALS system (Wyatt).

## Thermal stability assays

Thermal stability measurements were carried out using the Prometheus system (Nanotemper) which measures the effects of a range of pHs on thermal unfolding of proteins and their aggregation. Bis-Tris buffers ranging from 6 to 9 in 0.5 increments with a 150 mM NaCl and 0.5mM TCEP. Melting temperatures were plotted in Prism.

## Fluorescence anisotropy

Anisotropy was measured in a 3x3 mm quartz cuvette using a JASCO FP-8500 fluorescence spectrometer equipped with polarizers. To determine the affinity of Chl1 for ssDNA, 10 nM FAM-labelled DNA was titrated with Chl1 full length or MiniChl1 V2 solutions also containing 10 nM of FAM DNA. The fluorescence anisotropy was measured after each addition at 484nm/ 520 nm excitation and emission wavelength, with 10-nm band width. Experiments were performed in buffer, containing 50 mM HEPES pH8.0, 50 mM NaCl and 0.5 mM TCEP at 25˚C.

Data were fitted to a quadratic binding curve:

$$A_{meas} = A_L + (A_{PL} - A_L)*([P] + [L] + K_d - sqrt(([P] + [L] + K_d)\hat{}2 - 4*[P]*[L]))/(2*[L])$$

With $A_{meas}$, measured anisotropy, $A_L$, anisotropy of the free ligand, $A_{PL}$, Anisotropy of the protein-ligand complex, [P], total concentration of protein, [L], total concentration of the ligand, and $K_d$, equilibrium dissociation constant.

## Supporting information

**S1 Raw Images.**
(PDF)

**S1 File.**
(PDF)

## Acknowledgments

We wish to thank D. Benton, L. Calder and E. Punch for helpful dicussions on EM methods, and D. Frith, P. Faull and F. Ibrahim from the Crick Proteomics Platform for single-band ID experiments.

## Author Contributions

**Conceptualization:** Zuzana Hodáková, Martin R. Singleton.

**Data curation:** Zuzana Hodáková, Martin R. Singleton.

**Formal analysis:** Zuzana Hodáková.

**Funding acquisition:** Martin R. Singleton.

**Investigation:** Zuzana Hodáková, Simone Kunzelmann, Shahid Mehmood.

**Methodology:** Zuzana Hodáková, Andrea Nans, Shahid Mehmood, Ian Taylor, Frank Uhlmann.

**Project administration:** Martin R. Singleton.

**Resources:** Peter Cherepanov.

**Supervision:** Peter Cherepanov, Martin R. Singleton.

**Validation:** Martin R. Singleton.

**Writing – original draft:** Zuzana Hodáková, Martin R. Singleton.

**Writing – review & editing:** Zuzana Hodáková, Frank Uhlmann, Peter Cherepanov, Martin R. Singleton.

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
