## [Decision Letter · Decision Letter 0]

7 Apr 2021

PONE-D-21-05603

Structural characterisation of the Chl1 helicase

PLOS ONE

Dear Dr. Singleton,

Thank you for submitting your manuscript to PLOS ONE. After careful consideration, we feel that it has merit but does not fully meet PLOS ONE’s publication criteria as it currently stands. Therefore, we invite you to submit a revised version of the manuscript that addresses the points raised during the review process.

This study is an significant scientific advance to the understanding of the mechanism of Chl1 in yeast and fungi. The authors demonstrate some interesting new findings, albeit at low resolution, regarding the structure of C . thermophilum Chl1.

Publication is recommended after the following changes as noted by the three reviewers and the AE.

The required alterations are primarily textual in nature. All of the reviewers comments must be addressed in a point-by-point fashion indicating the changes made and the corresponding line numbers. In particular, consider the following important issues that will increase textual clarity:The rationale behind using the fungus Chaeotmium thermophilum Chl1 gene (instead of the S cerevisiae gene discussed in the introduction) needs to be explicitly described.  It’s relationship to the yeast gene needs to be discussed in terms of identity and function. In addition, for clarity, please include the organism in the title to the paper.Are there any means for providing better reproductions of the rather blurry E-M data shown in Figure2?Please place the cartoon describing the background regarding the Fe-S closer currently in the Supplementary Material into the main set of Figures as indicated by the Reviewer.Please note the inaccurate order of the panels in Figure 4.All reviewers and the AE believe that this study represents a scientific advance and all describe minor changes to the manuscript. No major differences of opinion were present between all reviews.The independent review of the AE lead to several minor changes that include the further description of  C. thermophilum for clarity for the non-expert reader and the preference of data that better represents the data in Figure 2B as noted above. In addition the following minor textual changes were found:  Introduction Line 57: Define CMG;  Line 66: in yeast: Has the process been elucidated in other organisms?Results;' Line 120: Despite extensive efforts, (add comma); Line 141: Change “predictions suggesting” to “predictions and suggesting”; Line 147: Change “to nucleotide-free” to “to the nucleotide-free”; Line 181-182: Explain meaning of “the C. thermophilum ? and DDX11 sequences”.  Are you comparing organisms or genes?

We look forward to receiving your revised manuscript.

Kind regards,

Arthur J. Lustig, PhD

Academic Editor

PLOS ONE

Journal Requirements:

5. Please ensure that you refer to Figure 7 in your text as, if accepted, production will need this reference to link the reader to the figure.

6. Please include captions for your Supporting Information files at the end of your manuscript, and update any in-text citations to match accordingly. Please see our Supporting Information guidelines for more information: http://journals.plos.org/plosone/s/supporting-information

Reviewers' comments:

Reviewer's Responses to Questions

**Comments to the Author**

1. Is the manuscript technically sound, and do the data support the conclusions?

Reviewer #1: Yes

Reviewer #2: Yes

Reviewer #3: Yes

2. Has the statistical analysis been performed appropriately and rigorously? 

Reviewer #1: Yes

Reviewer #2: Yes

Reviewer #3: Yes

3. Have the authors made all data underlying the findings in their manuscript fully available?

Reviewer #1: Yes

Reviewer #2: Yes

Reviewer #3: Yes

4. Is the manuscript presented in an intelligible fashion and written in standard English?

Reviewer #1: Yes

Reviewer #2: Yes

Reviewer #3: Yes

5. Review Comments to the Author

Reviewer #1: This study is a good-faith effort to obtain a structure of the Chaetomium thermophilum Chl1 helicase, a member of the XPD clade of 5’-3’ helicases, which play a role in genome integrity.

The authors apply biophysical methods, especially cryo-EM, to characterize Chl1 structurally.

Whereas the work is technically proficient (as expected from this consortium of distinguished structural biologists at the Crick Institute), the results comprise a quite modest advance in our understanding of Chl1 structure.

The cryo-EM model obtained at 7.7 Å resolution (generously estimated, as the authors cite) illuminates the shape of the protein but provides no atomic features. The structure resembles that of other XPD helicases. They provide biophysical evidence for an intact Fe-S cluster, as expected.

They purify an isolated version of the insert domain, and a stable/folded insert-less version of Chl1 MiniChl1.V2), but no structures of these entities emerge. Nor is there a biochemical characterization of the ATPase and helicase activities of insert-less Chl1 compared to the native Chl1 that might shed light on the contributions, if any, of the insert to helicase biochemistry.

Notwithstanding these reservations, the paper satisfies the PLOS ONE publication criteria.

Text errors:

Line 30: “alterarations”

Line 31: “a the”

Line 135: “suugests”

Line 136: ‘minimsing’

Reviewer #2: Critical Comments:

Last sentence of Abstract: “suggesting a the insert domain…” Something grammatically wrong here with inclusion of both “a” and “the”

Line 64: “role of Chl1/DDX1 in SCC…” It appears that DDX1 should be DDX11

The abstract should state that it is a 7.7 Angstrom cryo-EM structure which has been solved.

Line 251: The authors state that full-length Chl1 has very low affinity for DNA. Does this correspond to data not shown? Can these data be included in the Supp Data section?

Line 59-60: Previously published work demonstrated that Timeless stimulates DDX11 helicase activity. Relevant reference: Cali et al., Nucleic Acids Research (2016) 44(2):705-17.

Lines 82-84: “20 kDa domain inserted in helicase domain I (HDI).” Because authors make a number of comparison statements of Chl1 and the domains such as the insert in HD1 to other Fe-S helicases including XPD and FANCJ, it is advised that a cartoon alignment of Fe-S cluster helicases being discussed in this work (e.g.., DDX11, FANCJ, RTEL1, XPD) along with Chl1 be included as a figure that is present in the manuscript (not Supp Data). The inclusion of this figure (as a sole figure or a panel of a figure) will help to orient the reader and extrapolate information/insights from the current study to other Fe-S cluster helicases being discussed in the manuscript.

Reviewer #3: Hodáková et. al report a medium-resolution structure of Chl1, a helicase involved in chromosome segregation in eukaryotes and a member of the XPD family of proteins. The authors report several similarities and differences between the structure of Chl1 and other XPD members. First, the authors report that the arch domain of apo-Chl1 remains in the ‘open’ state despite other apo-XPD family members’ arch domains being in the ‘closed’ state. Second, Chl1 contains an ‘insert’ in its helicase domain that is not present in many other XPD family members. The authors then use HDX mass spectrometry to show that the Fe-S cluster in Chl1 is ordered while the helicase insert is disordered. After constructing various insert-deletion variants, the authors use fluorescence anisotropy to show that Chl1 variants lacking the insert domain bind DNA much more strongly than the wild-type protein, suggesting an auto-regulatory function of the insert domain.

While the function of the open/closed state of the arch domain is left an open question, the experiments focusing on the insert domain suggest an intriguing auto-regulatory mechanism. Although follow up experiments are clearly needed to further define the roles of the arch and insert domains and strengthen the authors’ arguments, the experiments presented support the preliminary conclusions drawn. A few minor issues are discussed below:

1) In the introduction and abstract, multiple acronyms are defined that are never used again (ex. “FPC”, “CIP”, “G4”, etc). These should be taken out unless they are used later in the text.

2) In line 72, there is one spelled out number (“one”) and one digit number (“3”). The authors should be consistent with this.

3) The panels in Figure 4 are out of order and referenced incorrectly in the text and multiple figure legends.

4) In Figure 4C and related text it would be helpful to include the sequence numbers of the insert.

5) Also, in Figure 4C, it is unclear what the gray bars are referring to. If they are not being included in the graph, why are they shown?

6) In line 234, “regions” is misspelled.

7) The sentence that begins with “In higher eukaryotes…” in line 540 should be in the main text, not in the figure legend.

8) The sentence that begins with “It is possible that the human…” in line 555 should also be in the main text and not in the figure legend.

6. PLOS authors have the option to publish the peer review history of their article (what does this mean?). If published, this will include your full peer review and any attached files.

Reviewer #1: No

Reviewer #2: No

Reviewer #3: No

---

## [Author Response · Author response to Decision Letter 0]

20 Apr 2021

Responses to Referees

(line numbers refer to revised, non-tracked manuscript)

General comments from AE:

The rationale behind using the fungus Chaeotmium thermophilum Chl1 gene (instead of the S cerevisiae gene discussed in the introduction) needs to be explicitly described. It’s relationship to the yeast gene needs to be discussed in terms of identity and function. In addition, for clarity, please include the organism in the title to the paper.

We have expanded on the rationale for using this Chl1 ortholog, and further explained its relationship to the yeast gene (lines 108-115). We have amended the title as requested.

Are there any means for providing better reproductions of the rather blurry E-M data shown in Figure2?

The blurriness of the EM images shown (especially in Fig. 2B) reflect the fundamentally limited resolution of the experimental data, so there is unfortunately no way we can improve on these.

Please place the cartoon describing the background regarding the Fe-S closer currently in the Supplementary Material into the main set of Figures as indicated by the Reviewer.

We have incorporated the cartoon illustration of the Chl1 domains including the Fe-S cluster into Figure 1.

Please note the inaccurate order of the panels in Figure 4.

This has been corrected in the figure legend.

Introduction Line 57: Define CMG; Line 66: in yeast: Has the process been elucidated in other organisms?

We have defined CMG in the text (line 57). The CMG is a conserved amongst eukaroytes, but the extent of functional conservation of Chl1 is unclear (lines 66-68).

Results;' Line 120: Despite extensive efforts, (add comma); Line 141: Change “predictions suggesting” to “predictions and suggesting”; Line 147: Change “to nucleotide-free” to “to the nucleotide-free”; Line 181-182: Explain meaning of “the C. thermophilum ? and DDX11 sequences”. Are you comparing organisms or genes?

All these have been corrected and the section on the insert sequences re-worded (now lines 189-190).

Reviewer #1 

Text errors:

Line 30: “alterarations”

Line 31: “a the”

Line 135: “suugests”

Line 136: ‘minimsing’

All text errors have been corrected.

Reviewer #2

Last sentence of Abstract: “suggesting a the insert domain…” Something grammatically wrong here with inclusion of both “a” and “the”

Corrected.

Line 64: “role of Chl1/DDX1 in SCC…” It appears that DDX1 should be DDX11

The abstract should state that it is a 7.7 Angstrom cryo-EM structure which has been solved.

Both corrected.

Line 251: The authors state that full-length Chl1 has very low affinity for DNA. Does this correspond to data not shown? Can these data be included in the Supp Data section?

This corresponds to the low affinity observed in our FP binding assay. The wording has been slightly altered to clarify this (line 260).

Line 59-60: Previously published work demonstrated that Timeless stimulates DDX11 helicase activity. Relevant reference: Cali et al., Nucleic Acids Research (2016) 44(2):705-17.

Reference included.

Lines 82-84: “20 kDa domain inserted in helicase domain I (HDI).” Because authors make a number of comparison statements of Chl1 and the domains such as the insert in HD1 to other Fe-S helicases including XPD and FANCJ, it is advised that a cartoon alignment of Fe-S cluster helicases being discussed in this work (e.g.., DDX11, FANCJ, RTEL1, XPD) along with Chl1 be included as a figure that is present in the manuscript (not Supp Data). The inclusion of this figure (as a sole figure or a panel of a figure) will help to orient the reader and extrapolate information/insights from the current study to other Fe-S cluster helicases being discussed in the manuscript.

We have included a cartoon schematic of the XPD family sequences and approximate structural arrangement in figure 1A, highlighting the domain organisation and Fe-S location.

Reviewer #3:

1) In the introduction and abstract, multiple acronyms are defined that are never used again (ex. “FPC”, “CIP”, “G4”, etc). These should be taken out unless they are used later in the text.

This has been done. “FPC” is used later in the revised text.

2) In line 72, there is one spelled out number (“one”) and one digit number (“3”). The authors should be consistent with this.

Corrected.

3) The panels in figure 4 are out of order and referenced incorrectly in the text and multiple figure legends.

Corrected.

4) In Figure 4C and related text it would be helpful to include the sequence numbers of the insert.

The sequence numbers of the insert are now shown in the figure and the legend.

5) Also, in Figure 4C, it is unclear what the gray bars are referring to. If they are not being included in the graph, why are they shown?

The gray bars show peptides that were detected in the MS experiments, but not shown on the graph because they represented areas of multiple coverage (for the sake of clarity, only single peptides are shown in the graph). As they add no useful data to the figure, we have removed them entirely.

6) In line 234, “regions” is misspelled.

Corrected.

7) The sentence that begins with “In higher eukaryotes…” in line 540 should be in the main text, not in the figure legend.

We have removed the sentence from the figure legend. The details of the inhibition by MAT1 are included in the discussion (lines 266-270).

8) The sentence that begins with “It is possible that the human…” in line 555 should also be in the main text and not in the figure legend.

Corrected.

---

## [Editor Report · Decision Letter 1]

23 Apr 2021

Structural characterisation of the  *Chaetomium thermophilum* Chl1 helicase

PONE-D-21-05603R1

Dear Dr. Singleton,

We’re pleased to inform you that your manuscript has been judged scientifically suitable for publication and will be formally accepted for publication once it meets all outstanding technical requirements.

Kind regards,

Arthur J. Lustig, PhD

Academic Editor

PLOS ONE
---

## [Editor Report · Acceptance letter]

29 Apr 2021

PONE-D-21-05603R1 

Structural characterisation of the *Chaetomium thermophilum* Chl1 helicase 

Dear Dr. Singleton:

I'm pleased to inform you that your manuscript has been deemed suitable for publication in PLOS ONE. Congratulations! Your manuscript is now with our production department. 

Kind regards, 

on behalf of

Dr. Arthur J. Lustig 

Academic Editor

PLOS ONE